# Experimental Study on Impact Friction Damage of Sweet Potato Skin

**Wanzhi Zhang [1,2], Yunzhen Qu [2], Xiubo Yin [3], Hongjuan Liu [4], Guizhi Mu [2,*] and Dengshan Li [2]**

[1] Collaborative Innovation Center for Shandong's Main Crop Production Equipment and Mechanization, Qingdao 266109, China; zwz@sdau.edu.cn

[2] College of Mechanical and Electrical Engineering, Shandong Agricultural University, Tai'an 271018, China

[3] Shandong General Station of Agricultural Technology Extension, Jinan 250100, China; 13181717070@163.com

[4] College of Agriculture, Shandong Agricultural University, Tai'an 271018, China

[*] Correspondence: muguizhi2008@163.com

**Abstract:** Sweet potato skin is prone to friction damage during mechanical harvesting. To reveal the friction damage mechanism of sweet potato skin, the impact friction process between a sweet potato and a rod was theoretically analyzed. The main factors affecting the impact friction force of the sweet potato skin include the sweet potato mass, drop height, distance from the center of the pendulum to the impact center of the sweet potato, maximum elastic displacement of the sweet potato, collision contact time, curvature radius of the sweet potato collision surface, material and roughness of the collision contact surface, etc. The mass, drop height, rod direction, rod state, and rod material of the sweet potato were used as the test factors, and the critical damage acceleration of the sweet potato skin was used as the test evaluation index. The results showed that the friction force caused by the collision between the sweet potato skin and rod increased with the increase in the sweet potato mass. The minimum friction is 3.5 N. The critical damage acceleration of the sweet potato skin decreased with the increase in the sweet potato mass, and the drop height had no significant effect on the critical damage acceleration of the sweet potato skin. Compared with the vertical placement, the critical damage acceleration of the sweet potato skin was smaller when the rod was placed horizontally, and the damage was more likely to occur. Under the same conditions, the critical damage acceleration of the sweet potato skin when the rod is rolling is greater than that when the rod is fixed. The critical damage acceleration of the impact friction between the sweet potato and 65Mn rod is the smallest, and the critical damage acceleration of the impact friction with the 65Mn–leather rod is the largest.

**Keywords:** sweet potato; collision damage; peel damage; acceleration; critical damage

## 1. Introduction

The broken skin of sweet potatoes is one of the main damage forms in the process of mechanized sweet potato harvesting [1–4]. The friction and collision between sweet potatoes and mechanical parts are the main causes of skin damage in the transmission process of the sweet potato excavator and the screening process of the potato soil separation device [5–7]. Sweet potato skin damage not only affects the appearance of sweet potatoes but also causes the overall quality of sweet potatoes to decline and affects the selling price of sweet potatoes. Due to the incomplete surface of the defect, it is easily attacked by bacteria, which accelerates the deterioration of sweet potato blocks and is not conducive to the storage of sweet potatoes [8–10].

At present, there is no report on the impact friction and damage of sweet potato skin which can be referred to in the relevant research on potatoes. Scholars at home and abroad have conducted a combined study on the damage of sweet potatoes and potato skin and the impact of damage on its internal tissues [11,12]. Geyer et al. [13] inserted an acceleration sensor into the potato and used the free-fall collision device to make the potato collide with the pressure sensor, and studied the effects of potato variety, mass, fall height, sensor

installation position, etc. on the collision acceleration and collision contact force. Rady et al. [14] used a free-fall collision device to test the respiration rate, damage area, and damage volume of a potato ball after collision with a steel plate and rod. Daniel et al. [15] developed a computer-controlled pendulum impact device that can quickly detect the impact acceleration and force of potatoes. Thomson et al. [16] studied the effects of potato mass, temperature, and sorting acceleration on potato skin damage during potato seed sorting. The larger the potato mass, the lower the temperature, and when the sorting acceleration exceeds a certain value, the potato skin damage will be aggravated. Alexei et al. [17] used data recording balls instead of potatoes to study the stress and damage of potatoes during the operation of three different models of potato harvesters. Gao Guohua et al. [18] designed a mechanical properties test and drop impact test of sweet potatoes, established the mathematical model relationship between the drop height, impact force, and impact stress, and finally measured the firmness of the sweet potatoes as 1.3385 MPa and the critical damage impact force as 424.2 N. Xie Shengshi et al. [19] designed a potato collision test platform to solve the problem of mechanical damage to potatoes during harvest. Through orthogonal test analysis, it was found that the significant factors affecting potato damage volume were initial height, potato mass, potato temperature, and collision material in sequence. M. Bentini et al. [20] studied the damage to potatoes during harvest, proposed that the impact of the separating parts of the harvester was the main cause of the damage to the potatoes, and revealed the source of the impact from the perspectives of the advancing speed of the machine and soil moisture. Bajema et al. [21] carried out a static impact test of potatoes. The results showed that the failure stress, strain, and elastic modulus of the potato decreased with the increase in temperature. P. Azizi et al. [22] studied potato motion in Visual NASTRAN software. The device conducted field tests under different forward speeds, leaf angles, and rotational speeds, and obtained the best parameter combination to control the potato damage rate within 4%.

According to the principle of a single pendulum impact scratch test, Deng Weigang made a theoretical analysis of the impact friction process of potato skin. The mathematical model of impact friction between potato and cylindrical rod is of great significance to this paper [23].

In this paper, a sweet potato skin impact friction test bench was built based on references to analyze the impact friction characteristics between sweet potatoes and the rod of the potato soil separation device and the relationship between the impact friction characteristics and the sweet potato skin damage and to obtain the relevant factors and rules affecting the impact friction acceleration of the sweet potatoes, so as to reveal the damage mechanism of sweet potato skin breaking and provide a basis for the low-damage optimization design of a mechanized sweet potato harvesting device.

## 2. Materials and Methods

### 2.1. Test Apparatus

When the sweet potato skin impinges on the rod, it is necessary to dynamically adjust the compression displacement of the contact site and the vertical direction between the sweet potato and the rod. For this reason, a single-pendulum impact friction test device was designed, as shown in Figure 1. The top of the support bracket of the test device and the end of the supporting shaft are each provided with a rectangular slot, and the two are connected by bolts. After loosening the adjusting nut, the support shaft can move back and forth in the horizontal direction and up and down in the vertical direction. The end of the support shaft is fitted with a rolling bearing and is fixed with a light pendulum rod. The sweet potato is held at the end of the light pendulum rod lifted to a certain height, and then released. When it swings to near the lowest position, it will collide with the rod.

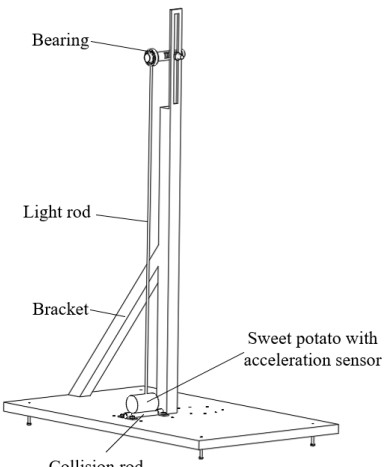

**Figure 1.** Sweet potato skin and cylindrical rod impact friction test equipment.

### 2.2. Impact Friction Analysis

The pendulum impact friction diagram between sweet potato skin and cylindrical rod is shown in Figure 2. Point P is the farthest point from the sweet potato surface to point $O_1$. When the sweet potato swings to the vertical position, point P moves to point A, and point A moves to point A′ due to elastic deformation caused by extrusion; the size of AA′ can be changed by adjusting the relative position of the sweet potato and the rod in the lowest position. The initial position of the collision between the sweet potato and the rod is $P_0$. According to the shape characteristics of the sweet potato surface, it can be assumed that $P_0$ and $P_1$ are on a part of the arc near the lowest point P. During the collision between the sweet potato and rod, the arc surface of the $P_0P$ section is the elastic compression stage, and the elastic deformation of the sweet potato gradually increases from zero. When the P point is in contact with the rod, the elastic deformation increases to the maximum. The elastic recovery stage is when the arc surface of $PP_1$ is in contact with the rod, and the $P_1$ point is the contact point when the sweet potato and the rod are out of collision, and the elastic deformation of the sweet potato is zero.

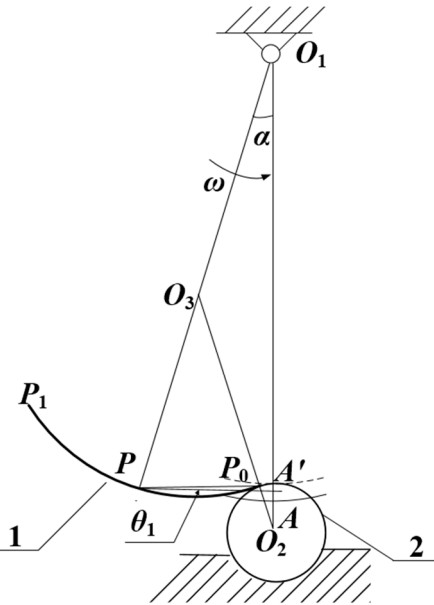

**Figure 2.** Schematic diagram of impact friction between sweet potato skin and cylindrical rod. 1. Local surface of sweet potato; 2. collision rod.

In Figure 2, the single pendulum angle $\alpha$ corresponding to the initial collision contact point $P_0$ is called the initial collision angle, the swing center of the single pendulum is $O_1$, the center of the cylindrical rod is $O_2$, and the center of curvature of the sweet potato surface at point P is $O_3$. When $\Delta O_1 O_2 O_3$,

$$
\begin{aligned}
O_1 O_2 &= L + R_1 - \lambda \\
O_1 O_3 &= L - R_2 \\
O_2 O_3 &= R_1 + R_2
\end{aligned}
\tag{1}
$$

where $L$ denotes the distance from the center of the pendulum to the lowest point $O_1 P$ of the sweet potato, mm; $R_1$ denotes the radius of curvature of the rod, mm; $R_2$ denotes the radius of curvature of the sweet potato surface at point A, mm; $\lambda$ denotes the maximum elastic deformation of sweet potato $AA'$, mm.

The initial angle of collision between the sweet potato and the rod can be solved according to the cosine law.

$$
\alpha = \arccos \left[ \frac{(L - R_2)^2 + (L + R_1 - \lambda)^2 - (R_1 + R_2)^2}{2(L - R_2)(L + R_1 - \lambda)} \right]
\tag{2}
$$

where $\alpha$ denotes the initial angle of collision between sweet potato and rod with the unit of rad.

According to the relationship between the angles of $\Delta O_1 O_2 O_3$ and $\Delta O_3 P P_0$,

$$
\theta_1 = \frac{1}{2}(\angle O_1 O_2 O_3 - \alpha) = \frac{1}{2} \left[ \arcsin(\frac{L - R_2}{R_1 + R_2} \sin \alpha) - \alpha \right]
\tag{3}
$$

where $\theta_1$ denotes the tilt angle between $P_0 P$ and the horizontal direction when the collision contact point is $P_0$ with the unit of rad.

### 2.2.1. Establishment of Collision Physical Model

According to the collision process between the sweet potato and the rod in Figure 2, it can be seen that the collision process starts from the contact point $P_0$ and reaches the maximum elastic displacement $\lambda$ at the point P. According to the principle of relative motion, it is assumed that the simple pendulum and the sweet potato are fixed in the vertical position, and the rod has impact friction with the surface of the sweet potato at a certain speed V. The camber $P_0 P$ is simplified to a light rod $P_0 P$ that can rotate around $P_0$, and the maximum displacement of P point in the vertical direction is $\lambda$. The impact friction physical model established is shown in Figure 3, where $k$ and $c$ represent the stiffness coefficient and damping coefficient of the sweet potato in the vertical direction, respectively.

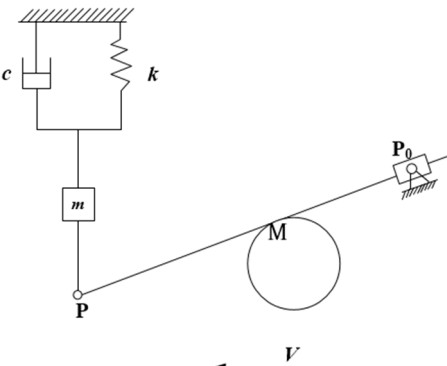

**Figure 3.** Physical model of impact friction between sweet potato and rod.

In the single-degree-of-freedom collision system shown in Figure 3, point P is between the displacement $y$ in the vertical direction and the exciting force $F$ in the vertical direction, satisfying the standard dynamic equation of the single-degree-of-freedom system.

$$m\ddot{y} + c\dot{y} + ky = F \tag{4}$$

where $m$ denotes the sweet potato mass with the unit of kg; $c$ denotes the system damping; $k$ denotes the system stiffness; $F$ denotes the vertical excitation force with unit of N.

The collision process from point $P_0$ to point P occurs in a very short time $\Delta t$, and the collision process satisfies the impulse theorem.

$$m\Delta V = F\Delta t \tag{5}$$

where $\Delta V$ denotes the change in vertical velocity of sweet potato during the collision with the unit of m/s; $\Delta t$ denotes the collision contact time from $P_0$ to P with the unit of s.

2.2.2. Solution of Friction $F_f$ on the Collision Surface of Sweet Potato

The impulse theorem is applied in the vertical direction for the collision process from the beginning of the collision to the maximum elastic displacement $\lambda$ [24].

$$F = \frac{m\Delta V}{\Delta t} = \frac{m\sqrt{2gH}\sin\theta_1}{\Delta t\cos^2\theta_1} \tag{6}$$

where $H$ denotes the height at which the sweet potato was raised before the collision began, $m$.

According to the collision physical model in Figure 3, ignoring the resistance caused by the slight movement of the light rod along the $PP_0$ direction, the friction force $F_f$ during the collision between the sweet potato and the rod is approximately equal to the magnitude of the component $F_2$ of the excitation force $F$ of point P in the vertical direction along the $PP_0$ direction of the rod, as shown in Figure 4.

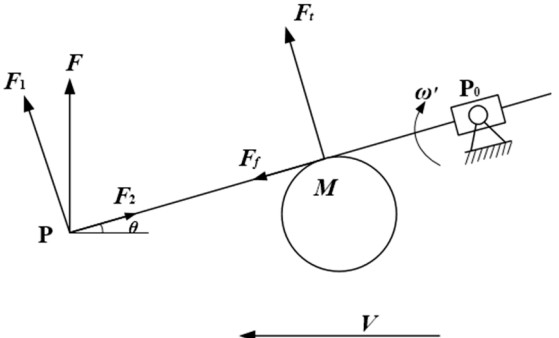

**Figure 4.** Force analysis between sweet potato and rod collision process. $\omega'$ (rad/s), the instantaneous angular velocity of light rod $P_0P$; $F_t$ (N), the collision force of the collision contact point along the vertical direction of $P_0P$; $F_f$ (N), the friction force of collision between sweet potato and rod; $F$ (N), the vertical excitation force at point P; $F_1$ (N), the component of the exciting force $F$ along the vertical rod $PP_0$ direction; $F_2$ (N), the component of the exciting force $F$ along the $PP_0$ direction of the rod.

Therefore,

$$F_f = F\sin\theta \tag{7}$$

where $\theta$ denotes the tilt angle between $P_0P$ and the horizontal direction when the collision contact point is M with the unit of rad; the variation range of $\theta$ is between $\theta_1$ and $\theta_2$, $\theta_2$ is the tilt angle between rod $P_0P$ and the horizontal direction when the collision contact point is P with the unit of rad. Since $\theta_1$ to $\theta_2$ changes very little, use $\theta_1$ instead of the size of $\theta$.

According to Formulas (6) and (7), the following can be obtained:

$$F_f = \frac{m\sqrt{2gH}}{\Delta t}\tan^2\theta_1 \tag{8}$$

According to the calculated expressions of $\alpha$ and $\theta_1$, the main factors affecting the magnitude of friction force $F_f$ are the distance $L$ from the lowest point of the sweet potato surface to the center of the pendulum, the radius of the cylindrical rod $R_1$, the radius of curvature $R_2$ at the lowest position of the sweet potato collision, the mass $m$ of the sweet potato, the height of the drop $H$, the maximum elastic displacement $\lambda$ and the collision contact time $\Delta t$ from the beginning of the collision to the production of the maximum elastic displacement $\lambda$.

*2.3. Test System*

The sweet potato impact friction test system includes the impact friction test device shown in Figure 1 and the acceleration acquisition system shown in Figure 5. The acceleration acquisition system mainly includes a data acquisition and analyzer, a 12 g-mass acceleration sensor with an accuracy of $\pm10$ mv/g and dimensions of $\Phi12$ mm $\times$ 21 mm (ZC1001L; Yangzhou Ketu electronic Co., Yangzhou, China), a signal conditioner and a computer.

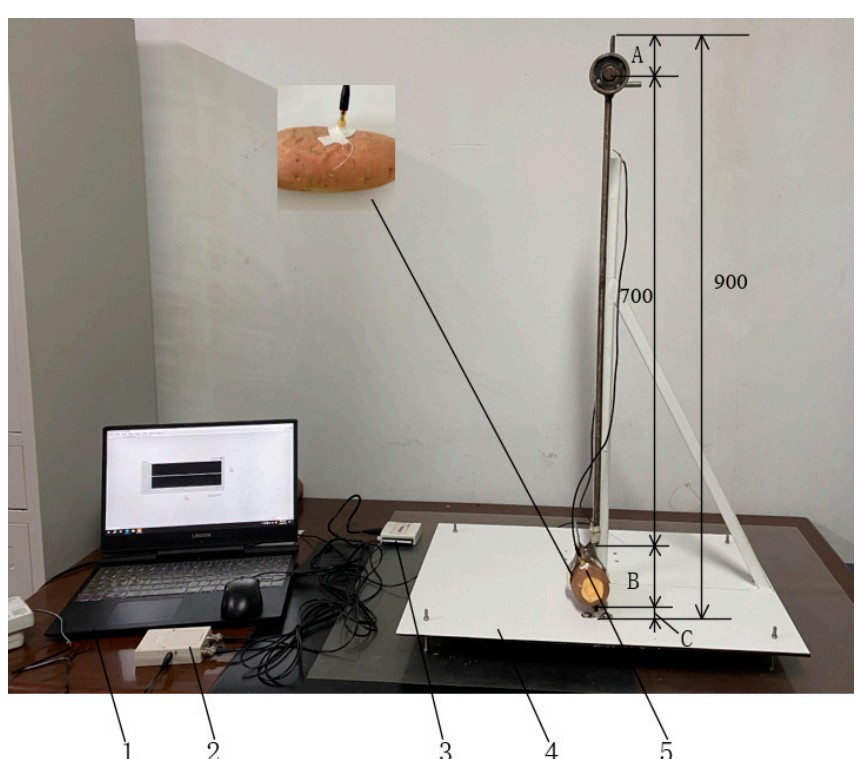

**Figure 5.** Sweet potato impact friction acceleration acquisition system. 1. Computer; 2. signal conditioner; 3. date acquisition card; 4. testbed; 5. acceleration sensor.

Before the test began, a knife was used to open a blind hole with a diameter of 12 mm and a depth of 25 mm on the surface of the sweet potato, and the sensor was wrapped with waterproof foam and inserted into the hole to fix it. The height between the top of the test stand and the surface of the test stand is 900 mm, and the height between the center of the support shaft and the top surface of the fixture is 700 mm. The distance $A$ from the top of the test stand to the center of the support shaft, the distance $B$ from the top surface of the fixture to the lowest position of the sweet potato, and the distance $C$ from the surface of the test stand to the highest position of the rod were respectively measured, as shown in

Figure 5. To make the potato and the rod produce mutual friction, the vertical position of the support axis was adjusted, so that the overlap distance between the sweet potato and the rod in the vertical direction was 1 mm~2 mm, and the parameters met the following formula.

$$A + B + 700 + C - 900 = 1 \sim 2 \tag{9}$$

*2.4. Test Scheme*

The sweet potato impact friction test scheme is shown in Table 1. The experimental factors were sweet potato quality, initial height, rod direction, rod state, and rod material. The test index was the critical damage acceleration of sweet potato skin damage. The rod is 65Mn with a diameter of 10 mm, and the plastic and leather with a thickness of 2 mm adhere to the surface of the rod, respectively, to simulate the collision between the sweet potato and the rod with different materials. The experiment was repeated 10 times in each group, and the average value of the maximum acceleration of sweet potato skin damage was taken as the test result.

**Table 1.** Impact friction test scheme of sweet potato.

| Number | Sweet Potato Mass (g) | Initial Altitude (mm) | Rod Direction | Rod Condition | Rod Material |
|---|---|---|---|---|---|
| 1 | 100 | | | | |
| 2 | 200 | | | | |
| 3 | 300 | 80 | Horizontal | Static | 65Mn |
| 4 | 400 | | | | |
| 5 | 500 | | | | |
| 6 | | 30 | | | |
| 7 | | 40 | | | |
| 8 | 250 | 50 | Horizontal | Static | 65Mn |
| 9 | | 60 | | | |
| 10 | | 70 | | | |
| 11 | 100 | | | | |
| 12 | 200 | | | | |
| 13 | 300 | 80 | Vertical | Static | 65Mn |
| 14 | 400 | | | | |
| 15 | 500 | | | | |
| 16 | 100 | | | | |
| 17 | 200 | | | | |
| 18 | 300 | 80 | Vertical | Rolling | 65Mn |
| 19 | 400 | | | | |
| 20 | 500 | | | | |
| 21 | 100 | | | | |
| 22 | 200 | | | | |
| 23 | 300 | 80 | Vertical | Static | 65Mn–leather |
| 24 | 400 | | | | |
| 25 | 500 | | | | |
| 26 | 100 | | | | |
| 27 | 200 | | | | |
| 28 | 300 | 80 | Vertical | Static | 65Mn–plastic |
| 29 | 400 | | | | |
| 30 | 500 | | | | |

## 3. Results and Discussion

*3.1. Impact Friction of Sweet Potato Skin*

According to the test results of the five groups of tests No. 11–15, the collision friction force was calculated by selecting the two test data of sweet potato skin damage in each group, and the results are shown in Table 2. The collision friction force Ff was calculated from the previous Formula (8), and the collision time Δ*t* was read from the signal collected

by the acceleration sensor. The curvature radius of a sweet potato was mapped by marking the impact friction point between the sweet potato skin and rod with a marker and cutting it with a knife after the collision. The section outline of the cut part was printed on white paper with a pencil to obtain its outline curve, and then the center of the section outline curve was made by a geometric drawing to measure the curvature radius $R_2$ of the collision contact area between the sweet potato and the rod. According to the formula of the friction force, the change in the friction force is mainly affected by the mass and collision time of the sweet potato. The difference in the surface curvature of the sweet potato with the same mass will lead to the difference in the collision time, and the friction force value will fluctuate greatly.

**Table 2.** Calculation results of skin impact friction of sweet potato.

| Number | Rod Radius $R_1$ (mm) | The Sweet Potato Curvature Radius $R_2$ (mm) | Maximum Elastic Displacement $\lambda$ (mm) | Pivot to Sweet Potato Lowest Point $L$ (mm) | Drop Height $H$ (mm) | Sweet Potato Mass $m$ (g) | Test Value $\Delta t$ (ms) | Friction Force $F_f$ (N) |
|---|---|---|---|---|---|---|---|---|
| 1 | 5 | 23 | 2 | 751 | 800 | 100 | 2.1 | 3.5 |
| 2 | 5 | 25 | 2 | 765 | 800 | 100 | 1.9 | 3.8 |
| 3 | 5 | 30 | 2 | 760 | 800 | 200 | 2.2 | 6.6 |
| 4 | 5 | 25 | 2 | 777 | 800 | 200 | 2.7 | 5.4 |
| 5 | 5 | 30 | 2 | 761 | 800 | 300 | 2 | 11 |
| 6 | 5 | 31 | 2 | 763 | 800 | 300 | 2.9 | 7.5 |
| 7 | 5 | 33 | 2 | 781 | 800 | 400 | 2.5 | 11.8 |
| 8 | 5 | 37 | 2 | 771 | 800 | 400 | 2 | 14.7 |
| 9 | 5 | 34 | 2 | 786 | 800 | 500 | 2.4 | 15.3 |
| 10 | 5 | 44 | 2 | 790 | 800 | 500 | 3.1 | 11.8 |

According to the results in Table 2, the minimum friction force causing skin damage to the sweet potato is 3.5 N, and the collision friction time $\Delta t$ between the sweet potato and the rod is concentrated in the range of 2 ms–3 ms. The overall results showed that the friction force increased gradually with the increase in the mass of the sweet potato.

*3.2. Analysis of Critical Damage Acceleration of Sweet Potato Skin*

The critical damage acceleration curve of sweet potato skin was obtained by using the test system in Figure 2, as shown in Figure 6. At the beginning of the collision, the collision acceleration of the sweet potato increases continuously and reaches the maximum value when it reaches the maximum elastic displacement. Because the sweet potato is fixed by the metal clip, the slight displacement up and down along the normal direction of the contact surface occurs continuously during the collision friction, so the collision friction acceleration fluctuates continuously within a certain range. Towards the end of the collision, the acceleration decreases and decreases until it reaches zero. After the end of each group of tests, damage to the sweet potato skin was observed. After the end of each group of experiments, the peak value of the acceleration curve of the sweet potato with skin damage (Figure 7) was selected as the critical acceleration of the sweet potato skin damage. To ensure the accuracy of the test results, the results of multiple tests in the same group were taken to calculate the average value.

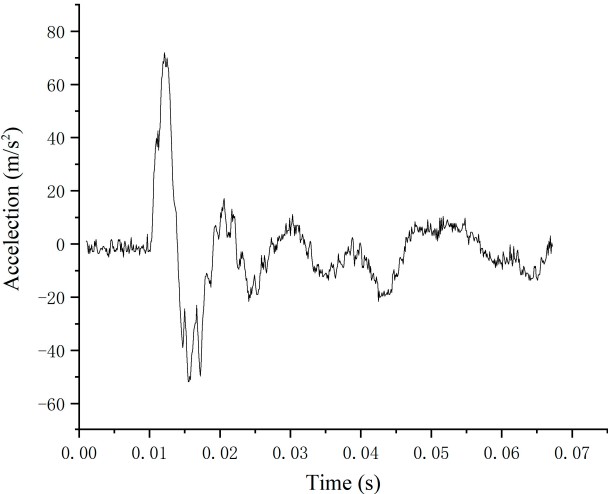

**Figure 6.** Critical damage acceleration curve of sweet potato skin.

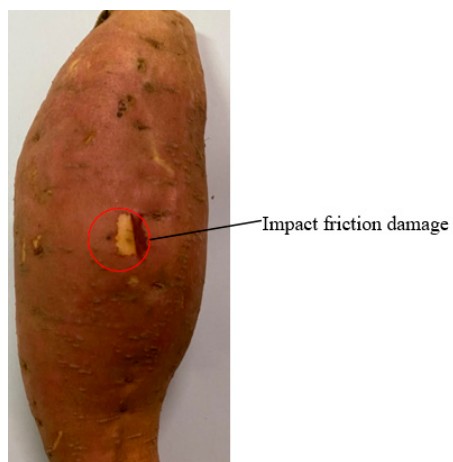

**Figure 7.** Sweet potato skin damage.

3.2.1. Effects of Sweet Potato Mass and Rod Direction on Critical Damage Acceleration of Sweet Potato Skin

According to the test scheme in Table 1 (test No. 1–5, 11–15), the impact friction test of the sweet potato skin was carried out, and the curve of the critical damage acceleration of the sweet potato and rod skin with the change in the sweet potato mass when the rod was fixed horizontally and vertically was drawn, as shown in Figure 8.

It can be seen from the figure that when the rod is fixed vertically or horizontally, the critical damage acceleration of the sweet potato skin decreases with the increase in mass. For the same mass of sweet potato, the collision of the sweet potato and rod in different directions has a great influence on the critical damage acceleration of the epidermis. The reason for this is that when the rod is placed horizontally, the collision area between the sweet potato and the rod is smaller, the collision time is longer, the friction force is smaller, and the acceleration is smaller. The fitting equations showed that the critical damage acceleration of the sweet potato skin was linearly related to the mass of the sweet potato when the rod was placed horizontally or vertically.

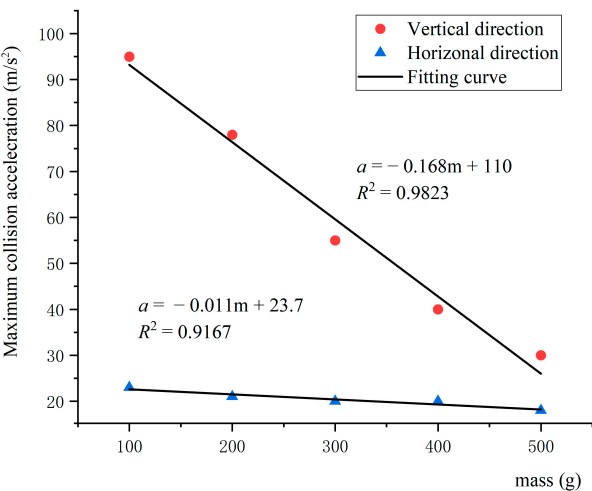

**Figure 8.** Relationship between critical damage acceleration of sweet potato skin and sweet potato mass.

### 3.2.2. Effect of Initial Height on Critical Damage Acceleration of Sweet Potato Skin

According to the test scheme in Table 1 (test No. 6–10), a sweet potato skin impact friction test was conducted, and the influence curve of the initial height on the friction acceleration of the sweet potato skin damage was drawn, as shown in Figure 9. It can be seen that when the rod is fixed horizontally, the correlation coefficient between the damage friction acceleration and the height is 0.192, indicating that there is no obvious correlation between the two, and the height has no significant effect on the critical damage acceleration of the sweet potato skin. The greater the initial height, the greater the initial velocity of the impact friction between the potato and rod, and the greater the impulse of the collision process. So the acceleration of the collision will be greater. The researchers analyzed the maximum impact of the potato on the rod. Similar conclusions are also obtained when considering the characteristics of velocity variation [14,25].

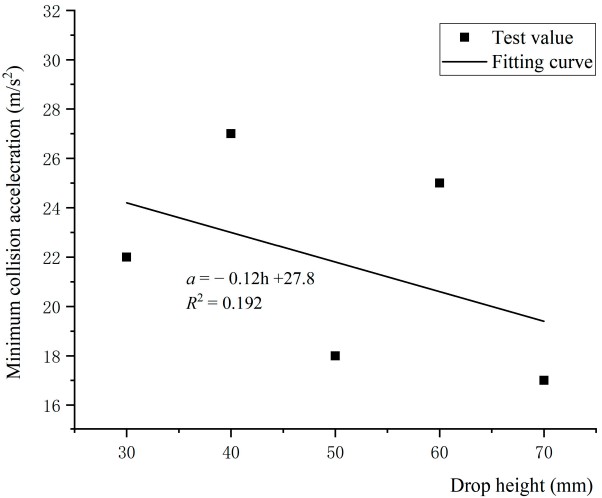

**Figure 9.** Relationship between critical damage acceleration and initial height of sweet potato skin.

### 3.2.3. Influence of Rod State on Critical Damage Acceleration of Sweet Potato Skin

According to the test scheme in Table 1 (test No. 11–20), the impact friction test of the sweet potato skin was carried out, and the curve of the influence of the sweet potato mass on the friction acceleration of the skin damage when the rod was fixed and rolled was obtained, as shown in Figure 10. For the same quality sweet potato, the critical damage acceleration of the skin when the rod was fixed was greater than that when the rod was

rolled. The fitting equations showed that the critical damage acceleration was linearly related to the mass of the sweet potato in both rod conditions.

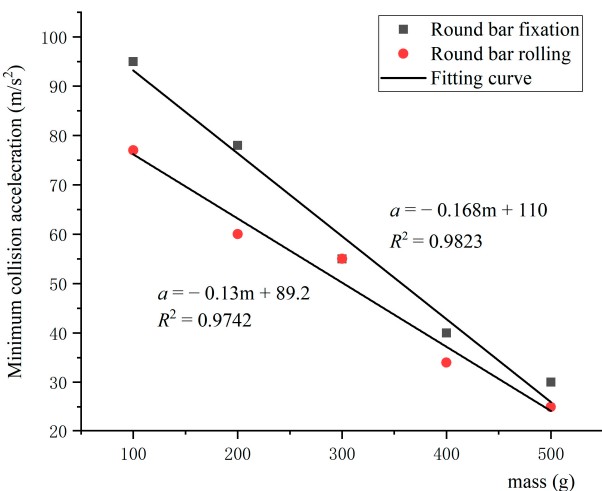

**Figure 10.** Relationship curves between critical damage acceleration and mass of sweet potato under different rod states.

When the sweet potato is in contact with the rod, the friction force will make the rod display a relative movement trend. When the rod is fixed, there is sliding friction between the sweet potato and the rod. When the two ends of the rod are fixed through the bearing seat, the rod is rotated around the rod axis by tangential force along the collision contact surface during the collision, and the friction between the sweet potato and the rod changes from sliding friction to rolling friction. Under the same condition as other collision conditions, the rolling friction force is less than the sliding friction force, so the critical damage acceleration of the skin when the rod is rolling is less than that when it is fixed.

### 3.2.4. Influence of Rod Material on Critical Damage Friction Acceleration of Sweet Potato Skin

According to the test scheme in Table 1 (test No. 11–15, 21–30), when the rod was fixed vertically, the influence curve of the rod material on the critical damage acceleration of the sweet potato skin was shown in Figure 11. The fitting equations showed that the three materials were consistent with the linear relationship between mass and critical damage acceleration, and the critical damage acceleration decreased with the increase in the mass of the sweet potato. When the sweet potato collided with a 65Mn–leather rod, the epidermal damage acceleration was the highest, followed by the collision with a 65Mn–plastic rod, and the damage acceleration with the 65Mn rod was the least. Xie Shengshi et al. [19] found that compared with the 65Mn and 65Mn–plastic, the damage area of the potato and 65Mn–rubber collision was the smallest, but the collision acceleration was the largest. This research result is similar to the analysis conclusion in this section.

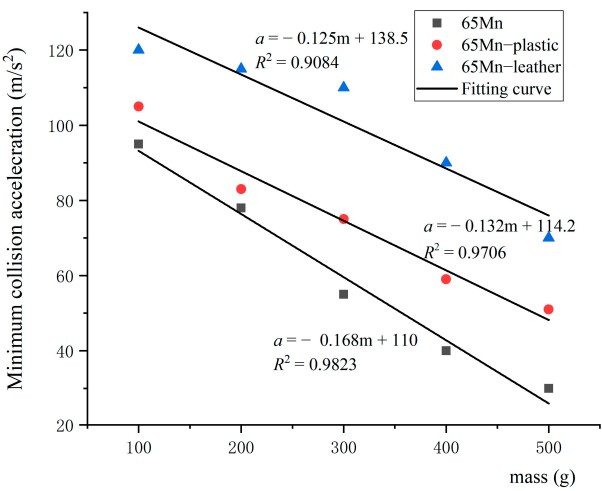

**Figure 11.** Influence curve of rod material on critical damage acceleration of skin.

### 3.3. Regression Analysis of Test Factors and Critical Damage Friction Acceleration of Sweet Potato Skin

According to the results of the test scheme in Table 1, linear regression analysis was performed on each test factor and the critical damage acceleration $a_t$ of the sweet potato skin is shown in Table 3. The results of the regression analysis showed that there was a significant linear negative correlation between the mass of the sweet potato and the critical damage acceleration of the sweet potato skin, and the correlation coefficient was above 0.9. The drop height had no significant effect on the critical injury acceleration of the sweet potato epidermis. The linear regression equation provides an effective prediction model for the acceleration of skin damage in the process of impact friction between sweet potatoes and different rods.

**Table 3.** Regression analysis of critical damage acceleration of sweet potato skin and experimental factors.

| Experimental Factor | Rod Parameter | Regression Equation | Correlation Coefficient ($R^2$) |
|---|---|---|---|
| Sweet potato quality (g) | Horizontal-Static-65Mn | a = −0.011m + 23.7 | 0.9167 |
| Drop height (mm) | Horizontal-Static-65Mn | a = −0.12h + 27.8 | 0.192 |
| Sweet potato quality (g) | Vertical-Static-65Mn | a = −0.168m + 110 | 0.983 |
| Sweet potato quality (g) | Vertical-Rolling-65Mn | a = −0.13m + 89.2 | 0.974 |
| Sweet potato quality (g) | Vertical-Static-65Mn–leather | a = −0.125m + 138.5 | 0.908 |
| Sweet potato quality (g) | Vertical-Static-65Mn–plastic | a = −0.132m + 114.2 | 0.971 |

## 4. Conclusions

The test results show that the friction force of the breakage between the sweet potato and rod increases gradually with the increase in the mass of the sweet potato. The effects of the different rod states on the critical damage acceleration of the sweet potatoes were compared and analyzed. When the rod was fixed horizontally and vertically, the critical damage friction acceleration decreased with the increase in the mass of the sweet potato. Under the same conditions, the critical damage friction acceleration when the rod is fixed in the horizontal direction is significantly smaller than that in the vertical direction, the critical damage friction acceleration when the rod is rolling is smaller than that when the rod is fixed, the critical damage friction acceleration when the sweet potato collides with the 65Mn rod is the smallest, and the critical damage friction acceleration when the sweet potato collides with the 65Mn–leather rod is the largest. The initial drop height had no significant effect on the critical acceleration of skin damage of sweet potatoes. In conclusion,

this study can provide a reference for the design of a chain rod sweet potato harvester to reduce the friction damage between sweet potatoes and rods during harvesting. In the future, the energy loss during the impact friction between the sweet potato and rod should be further studied.

**Author Contributions:** Conceptualization, G.M., Y.Q. and W.Z.; methodology, G.M., Y.Q., W.Z., X.Y., H.L. and D.L.; software, G.M. and Y.Q.; validation, G.M., Y.Q. and W.Z.; resources, H.L. and D.L.; data curation, Y.Q.; writing—original draft preparation, G.M. and Y.Q.; writing—review and editing, W.Z. and Y.Q. All authors have read and agreed to the published version of the manuscript.

**Funding:** This work was supported by the Collaborative Innovation Center for Shandong's Main crop Production Equipment and Mechanization (NO. SDXTZX-06) and Shandong Province Potato Industry Technology System Agricultural Machinery Post Expert Project, China (No. SDAIT-16-10).

**Institutional Review Board Statement:** Not applicable.

**Informed Consent Statement:** Not applicable.

**Data Availability Statement:** Data are contained within the article.

**Acknowledgments:** The authors thank the College of Mechanical and Electrical Engineering of Shandong Agricultural University and Agricultural Machinery Post Expert Project, China, for their facilities and support.

**Conflicts of Interest:** The authors declare that they have no known competing financial interests or personal relationships that could have appeared to influence the work reported in this paper.

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
