# Peer review of "Experimental Study on Impact Friction Damage of Sweet Potato Skin"

_agriculture, doi:10.3390/agriculture14010093_

Round 1
Reviewer 1 Report
Comments and Suggestions for Authors
To the best of my understanding of the submitted manuscript, the manuscript is partly within the scope of this journal. The authors presented impact friction damage of sweet potato skin during mechanical harvesting/transportation. The friction damage was theoretically presented. The English language of the submitted paper is satisfactory, and readers will understand the information presented. However, to maintain the impact and quality of the Journal, the Authors should revise their work according to the following comments:
1. Abstract and Conclusions: Lines 28-29 and Lines 330-334: What is the difference between 65Mn rod and 65Mn-leather?
2. Keywords: impact friction should be removed since it is part of the title. Provide 5 keywords different from the title.
3. Introduction: Lines 45-69: The citations given does not follow the Journal's Instructions for Authors. For e.g., Geyer in Line 45 should be Geyer et al. [13]. Correct citations format throughout the Introduction and the text accordingly.
- Objectives: Lines 66-69: The objective of the study is not clearly stated. Provide the clarity of the objectives of the study.
4. Materials and Methods: Table 3 format should be similar to Tables 1 and 2.
5. Results and Discussion: Line 227: what is 2ms-3ms....what is ms? Is ms the unit of time or collision friction time?
6. Figure 6: Time (ms): The SI unit of Time is not ms?
7. Figure 8: quality (g): How can you measure 'quality' of what in grams? Check also Figures 10, 11
8. Figure 9: Give a space between drop height and mm.
- There were no citations provided in the results and discussion. The discussion should also incorporate related published works.
9. References: The format should follow the Journals Instructions for Authors.
General Comment
The manuscript has some potential of scientific merit. Moderate English editing is required throughout the text/paper for clarity of the information presented. Adequate references/citations should be provided in the Introduction and results and discussion section to adequately meet scientific standard. The 21 references in the entire paper/manuscript are not adequate.
Comments on the Quality of English LanguageModerate English editing is required throughout the text/paper for clarity of the information presented.
Reviewer 2 Report
Comments and Suggestions for Authors
Are you considering deepening the study of the potato core and observing its behavior over time after collisions with the working organs of the machines that process and sort them?
See the comments made in the body of the paper!

Author Response
Thank you very much for taking the time to review this manuscript. We will then consider the internal effects of the sweet potato colliding with the rod of the harvester over time. For Figure 3, it is mentioned above that because the relative motion assumes that the sweet potato and the pendulum rod are fixed, the rod moves, so the rod moves at a certain speed V. The addition of the conclusion indicates that the study was aimed at reducing the skin bruising of sweet potatoes during harvesting. Other formatting issues have been modified in the paper.
